# A chemical catalyst enabling histone acylation with endogenous acyl-CoA

Misuzu Habazaki[1], Shinsuke Mizumoto[1], Hidetoshi Kajino[1], Tomoya Kujirai[2], Hitoshi Kurumizaka [2], Shigehiro A. Kawashima[1] ✉, Kenzo Yamatsugu[1] ✉ & Motomu Kanai [1] ✉

Life emerges from a network of biomolecules and chemical reactions catalyzed by enzymes. As enzyme abnormalities are often connected to various diseases, a chemical catalyst promoting physiologically important intracellular reactions in place of malfunctional endogenous enzymes would have great utility in understanding and treating diseases. However, research into such small-molecule chemical enzyme surrogates remains limited, due to difficulties in developing a reactive catalyst capable of activating inert cellular metabolites present at low concentrations. Herein, we report a small-molecule catalyst, *m*BnA, as a surrogate for a histone acetyltransferase. A hydroxamic acid moiety of suitable electronic characteristics at the catalytic site, paired with a thiol-thioester exchange process, enables *m*BnA to activate endogenous acyl-CoAs present in low concentrations and promote histone lysine acylations in living cells without the addition of exogenous acyl donors. An enzyme surrogate utilizing cellular metabolites will be a unique tool for elucidation of and synthetic intervention in the chemistry of life and disease.

Proper regulation of the chemical networks in life that determine the activity, localization, turnover, and interactions of biomolecules is key to maintaining homeostasis[1]. For example, acetylation of histone protein lysine residues is one of the most important protein post-translational modifications (PTMs) and it plays a pivotal role in epigenetic gene regulation by inducing structural changes in chromatin or recruiting reader proteins[2,3]. Dysregulation of epigenetic PTMs leads to disease[4,5]. Therefore, a method for intervening in the chemical network of epigenetic regulation has potential as both a therapy and as a way to elucidate functions of the epigenome.

Most of the epigenetic histone PTMs in cells are introduced by enzymes catalytically activating inert cellular metabolites as reagents. For example, histone acetyltransferases (HATs) promote acetylation of histone lysine residues using the endogenous thioester acetyl reagent acetyl-CoA (Ac-CoA) and regulate the histone acetylation level in coordination with histone deacetylases (HDACs). A traditional approach to intervening in the epigenetic chemical network is to use

small molecule inhibitors[6,7], activators[8,9], or recruiters[10,11] of enzymes to modulate the catalytic activity or the target-gene selectivity of the enzymes (Fig. 1a). This approach relies on functional endogenous enzymes. Therefore, if the HATs are genetically defective, for example, HDAC inhibitors cannot enhance acetylation levels[12]. Furthermore, methods using these molecules are vulnerable to the organism acquiring resistance[13,14].

We previously developed the chemical catalyst DSH to promote epigenetic histone lysine acetylation in living cells without reliance on enzymes[15,16]. DSH can use Ac-CoA as an acetyl source for lysine acetylation in test tubes. However, in-cell histone acetylation by DSH using endogenous Ac-CoA did not proceed, and the addition of a cell-permeable exogenous acetyl donor (NAC-Ac) in concentrations of 10 mM and greater was necessary for high-yielding histone acetylation (Fig. 1b). This suggested it would be difficult to chemically activate endogenous Ac-CoA, due to its low, micromolar-range concentration[17,18]. Currently, there is no chemical catalyst that can

[1]Graduate School of Pharmaceutical Sciences, The University of Tokyo, 7-3-1 Hongo, Bunkyo-ku, Tokyo 113-0033, Japan. [2]Institute for Quantitative Biosciences, The University of Tokyo, 1-1-1 Yayoi, Bunkyo-ku, Tokyo 113-0032, Japan. ✉e-mail: skawashima@mol.f.u-tokyo.ac.jp; yamatsugu@mol.f.u-tokyo.ac.jp; kanai@mol.f.u-tokyo.ac.jp

**a    Chemical modulator of enzyme activity**

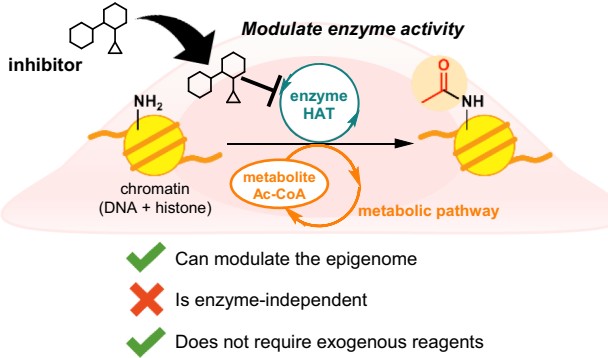

**b    Chemical catalyst & exogenous reagents promoting reaction**

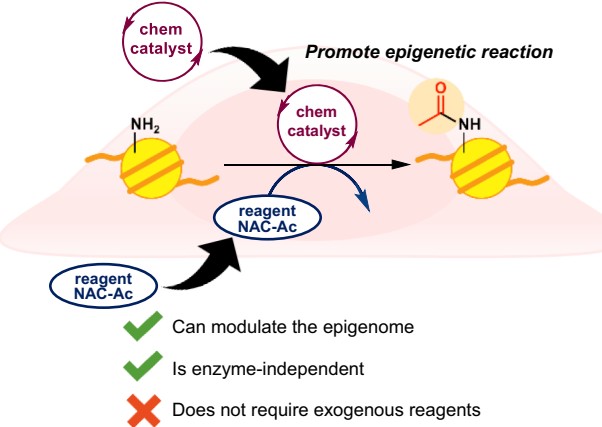

**c    This work: Chemical surrogate of enzyme activity**

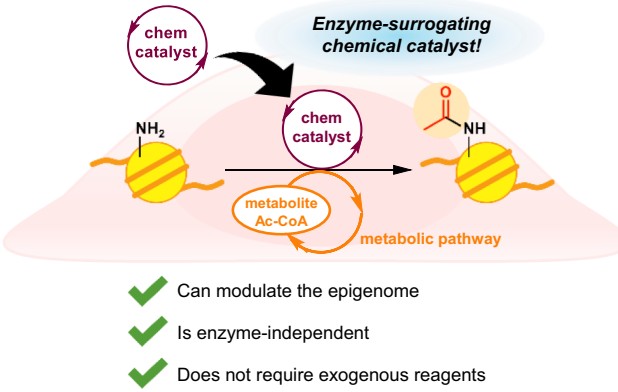

**Fig. 1 | Strategies for intervening in the epigenetic chemical network in cells.** **a** Modulating enzyme activity using small molecule mediators of enzymes. **b** Promoting epigenetic histone lysine acetylation using a chemical catalyst and an exogenous chemical reagent in live cells[15]. **c** A chemical surrogate for a HAT enzyme utilizing only cellular metabolites to promote histone lysine acylation in live cells, reported in this work.

promote physiologically important PTMs such as histone acetylation by filling the role of an enzyme and using solely cellular metabolites as reagents, despite the fact that such chemical catalysts would be useful for elucidating, modifying, and repairing dysregulated epigenetic reaction networks in cells.

Here we report *m*BnA, the chemical catalyst to promote epigenetically important histone lysine acylation in live cells by activating endogenous acyl-CoAs as the only acyl donors. The catalyst serves as both a chemical probe capable of detecting changes in nuclear-cytosolic acyl-CoA levels and may spur to the development of a small-

molecule surrogate for HAT enzymes enabling synthetic intervention in the cellular epigenome (Fig. 1c).

## Results

### Catalyst design for activating low-concentration Ac-CoA

To enable the use of endogenous Ac-CoA, we began the development of an acylation catalyst based on the reaction mechanism of DSH. DSH promoted the acetyl transfer reaction from thioester acetyl donors, including Ac-CoA, to a histone lysine residue through three main steps (Fig. 2a): (1) thiol-thioester exchange, (2) acetyl pyridinium formation via *S*-to-*N* acetyl migration, and (3) the subsequent acetyl transfer from *N*-acetyl pyridinium to a target lysine residue[16]. Kinetic studies revealed that the thiol-thioester exchange (1) was faster than steps (2) and (3)[19]. Assuming that this kinetic balance is the same under the conditions of low-concentration Ac-CoA, catalysts with facilitated *S*-to-*X* acetyl migration (and subsequent *X*-to-lysine *N* acetyl transfer) could promote the reaction irrespective of the concentration of Ac-CoA in the medium. DMAP, the nucleophilic core of DSH, is basic and thus is mostly deactivated by protonation under physiological conditions (i.e., neutral aqueous conditions)[20]. We previously reported that the piperidine-conjugated hydroxamic acid Ph-HXA (Supplementary Fig. 1a) is deprotonated under physiological conditions and serves as a nucleophilic catalyst superior to DMAP[21,22]. Furthermore, the acylation-active intermediate for Ph-HXA is electronically neutral, whereas the active intermediate for DSH is a cationic *N*-acyl pyridinium, the hydrolytic decomposition of which is expected to be non-negligible, especially when concentrations of acyl donors and acylation-active species are low. Based on these considerations, we designed a chemical catalyst, *p*HXA (Fig. 2b), which contains a hydroxamic acid moiety as a nucleophilic center to form a putative active *O*-acetylated hydroxamate species, and a thiol group as a handle to capture an acetyl group from Ac-CoA to form an *S*-acetylated intermediate (Supplementary Fig. 1b).

### Structural optimization of histone acetylation catalysts

The histone acetylation capabilities of *p*HXA catalysts were tested in test tubes. To enable the catalyst to target histone proteins, we used the LANA (latency-associated nuclear antigen of Kaposi's sarcoma-associated herpesvirus) peptide-inserted eDHFR (LieD) system[23]. In this system, the LieD protein binds with a catalyst-trimethoprim (TMP) conjugate and delivers the catalyst to the acidic patch of the nucleosome, promoting acetylation of the proximal histone lysine residue H2BK120 (Supplementary Fig. 2). *p*HXA-TMP conjugate **1** (the thiol group is masked as a disulfide, which is cleaved in-situ under reducing conditions[24], Fig. 3a), purified LieD protein, and Ac-CoA were incubated with recombinant nucleosome in a test tube, and then acetylation of H2BK120 was analyzed by both western blot using an anti-H2BK120ac antibody and liquid chromatography tandem mass spectrometry (LC−MS/MS) (Fig. 3b). The results indicate *p*HXA **1** was able to promote H2BK120 acetylation using Ac-CoA, but only in low yield (11%). The low histone acetylation activity of *p*HXA **1** prompted us to improve its catalytic activity through structural optimization.

We hypothesized that the low activity of the *p*HXA catalyst is due to moderate efficiency in generating the putative active *O*-acetylated species through *S*-to-*O* acetyl migration. The electron-withdrawing carbonyl group directly connected to the aromatic ring diminishes the electron density of the oxygen atom in *p*HXA, shifting the balance of the acetyl migration equilibrium in favor of the *S*-acetyl form (Fig. 2b). We therefore designed *m*HXA **2**, which has a carbonyl group at the 3-position of the aromatic ring, and *m*BnA **3**, which has an additional methylene group inserted between the aromatic ring and the amide carbonyl group, to reduce the acidity of the hydroxamic acid (Fig. 3a). Comparison of their histone H2BK120 acetylation activity in test tube reactions revealed that catalysts **2** and **3** are indeed more active than

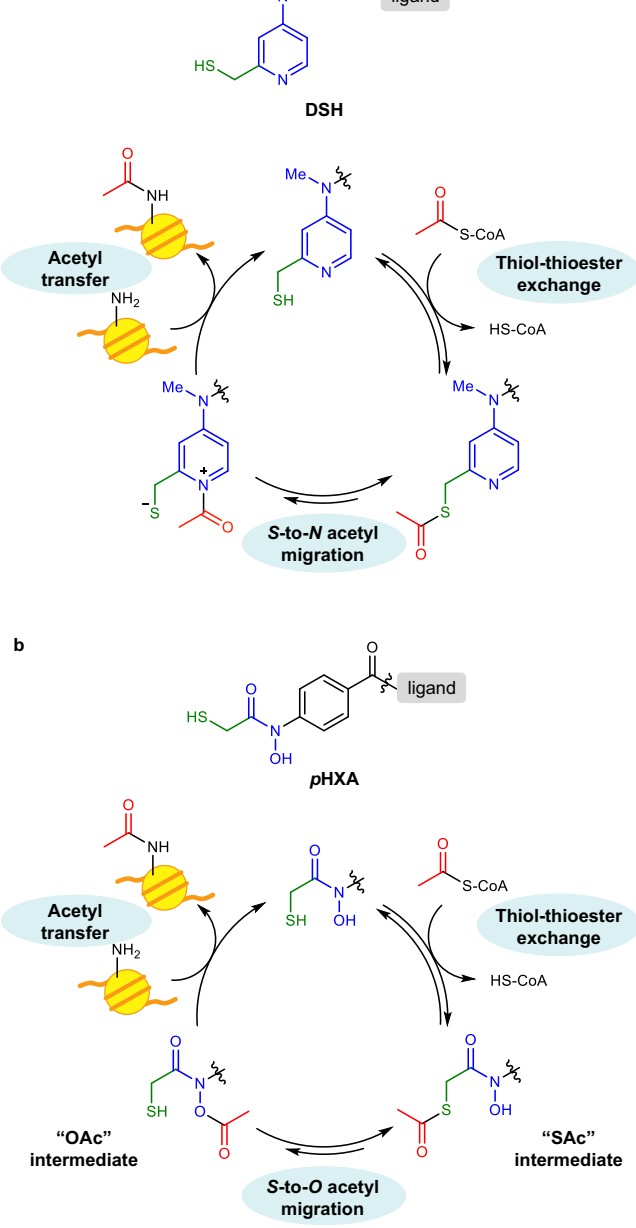

**Fig. 2 | Design of the hydroxamic acid-thiol conjugate catalyst *p*HXA. a** The structure of and a plausible reaction mechanism for DSH previously developed as a histone lysine acetylation catalyst[16]. **b** The structure of and a plausible reaction mechanism for the hydroxamic acid-thiol conjugate catalyst *p*HXA, designed for activating low-concentration Ac-CoA in cells.

catalyst **1**, with the highest acetylation yield by *m*BnA catalyst **3** (27%, Fig. 3b). Using ultraviolet-visible absorption (UV-Vis) spectroscopy, we determined the p$K_a$ values of *p*HXA and *m*BnA to be 8.3 and 8.6, respectively, supporting the hypothesis that the lower acidity of the hydroxamic acid is favorable for effective acetylation (Supplementary Fig. 3a–b).

To confirm our hypothesis that the low acidity of hydroxamic acid contributes to the improved acetylation activity by increasing the amount of the *O*-acetylated putative active species produced, we quantified the formation of the *S*- and *O*-acetylated intermediates from the catalysts. Catalysts **1–3** (25 μM) and Ac-CoA (5 mM) were mixed in Tris buffer (pH 7.5) in the presence of tris(2-carboxyethyl)phosphine hydrochloride (TCEP, 200 μM), and the formation of the *S*- and *O*-

acetylated catalysts was monitored with high-performance liquid chromatography (HPLC) and liquid chromatography-mass spectrometry (LC/MS) (Fig. 3c and Supplementary Fig. 4). As a result, the *S*-acetylated intermediates of all catalysts were smoothly and sufficiently formed to reach the equilibrium of thiol-thioester exchange within 2 h in almost the same yields. The concentration of the putative active *O*-acetylated intermediate was, however, about 4 times higher in *m*BnA **3** (9.1% in 3 h) than in *p*HXA **1** (2.0% in 3 h) and *m*HXA **2** (2.5% in 3 h). This tendency was consistent with the activity order in the H2BK120 acetylation reaction (Fig. 3b).

Shortening the linker between TMP and the *m*BnA catalyst further improved the H2BK120 acetylation yield to afford the optimized catalyst *m*BnA-TMP(gly$_1$) **4** (44% yield, Fig. 3a, b). Negative control catalysts, *O*-methylated and *S*-methylated derivatives of *m*BnA-TMP(gly$_1$) (**5, 6**: Fig. 3a) did not promote H2BK120 acetylation as efficiently as **4**, indicating that both the hydroxamic acid moiety and the thiol group were essential for the acetylation ability of the *m*BnA catalyst (18% and 11% for **5** and **6**, respectively, Supplementary Fig. 5), as expected from the postulated reaction mechanism (Fig. 2b).

### In-cell histone H2BK120 acetylation by *m*BnA catalyst

With the optimized catalyst *m*BnA-TMP(gly$_1$) **4** in hand, we next investigated in-cell acetylation of histone H2BK120. The concentration of endogenous Ac-CoA in cells is reported to be around 3–50 μM[17,18]. *m*BnA-TMP **4** promoted H2BK120-selective histone acetylation in test tubes with <50 μM Ac-CoA, suggesting that **4** can activate endogenous Ac-CoA and promote histone acetylation in living cells (Supplementary Fig. 6). LieD-transfected HEK293T cells were treated with catalysts alone (100 μM) for 10 h, and then histone acetylation levels were analyzed by western blot (Fig. 4a). DSH-TMP **S3**[16], Ph-HXA-TMP **S4**[21], and YZ-TMP **S5**[25], possessing previously reported catalyst centers, (Supplementary Fig. 7) failed to promote acetylation of H2BK120 in living cells without exogenous acetyl donors (lanes 1–3 and 9). The optimized catalyst *m*BnA-TMP **4**, however, promoted H2BK120 acetylation (2.5% yield, lane 5). The acetylation efficiency of *p*HXA-TMP **1** and *O*- and *S*-methylated *m*BnA-TMP catalysts (**5, 6**) were much lower than that of **4** (lanes 4–7), which is consistent with the results in test tubes (Fig. 3b and Supplementary Fig. 5). H2BK120 acetylation by *m*BnA-TMP **4** showed a positive correlation with both catalyst-treatment time and catalyst concentration (Supplementary Fig. 8).

The promotion of H2BK120 acetylation by *m*BnA-TMP **4** was suppressed by the addition of TMP as a binding competitor (lanes 5 and 8), indicating that the acetylation was due to proximity effects of the catalyst. Acetylation levels at histone lysine residues other than those proximal to the LANA binding site did not change by treatment with *m*BnA-TMP **4** (Supplementary Fig. 9). Furthermore, **4** promoted acetylation of H2BK120 when H2BK120ac writer enzymes p300 (KAT3B) and CBP (KAT3A)[26] were knocked down with RNAi (Supplementary Fig. 10). These results indicate that the promotion of H2BK120 acetylation by the *m*BnA catalyst was not dependent on cellular enzymatic activity, but was caused by the chemical catalyst itself in cells.

Western blot analysis of the whole-cell extract demonstrated that *m*BnA-TMP-mediated acylation was histone-selective (Supplementary Fig. 11). We also conducted chromatin immunoprecipitation (ChIP) analysis of H2BK120ac to determine genomic loci-dependency of *m*BnA-TMP-mediated acetylation. As a result, treatment of the cells by *m*BnA-TMP **4** increased H2BK120 acetylation levels in euchromatic and heterochromatic genes almost equally (Supplementary Fig. 12), indicating that **4** is able to promote acetylation irrespective of the chromatin structure or transcriptional state.

We compared the acetylation activity of **4** with p300, which is reported to target H2BK120[26] (Supplementary Fig. 13). Overexpression of p300 increased the acetylation level of H3K18[27,28], while that of

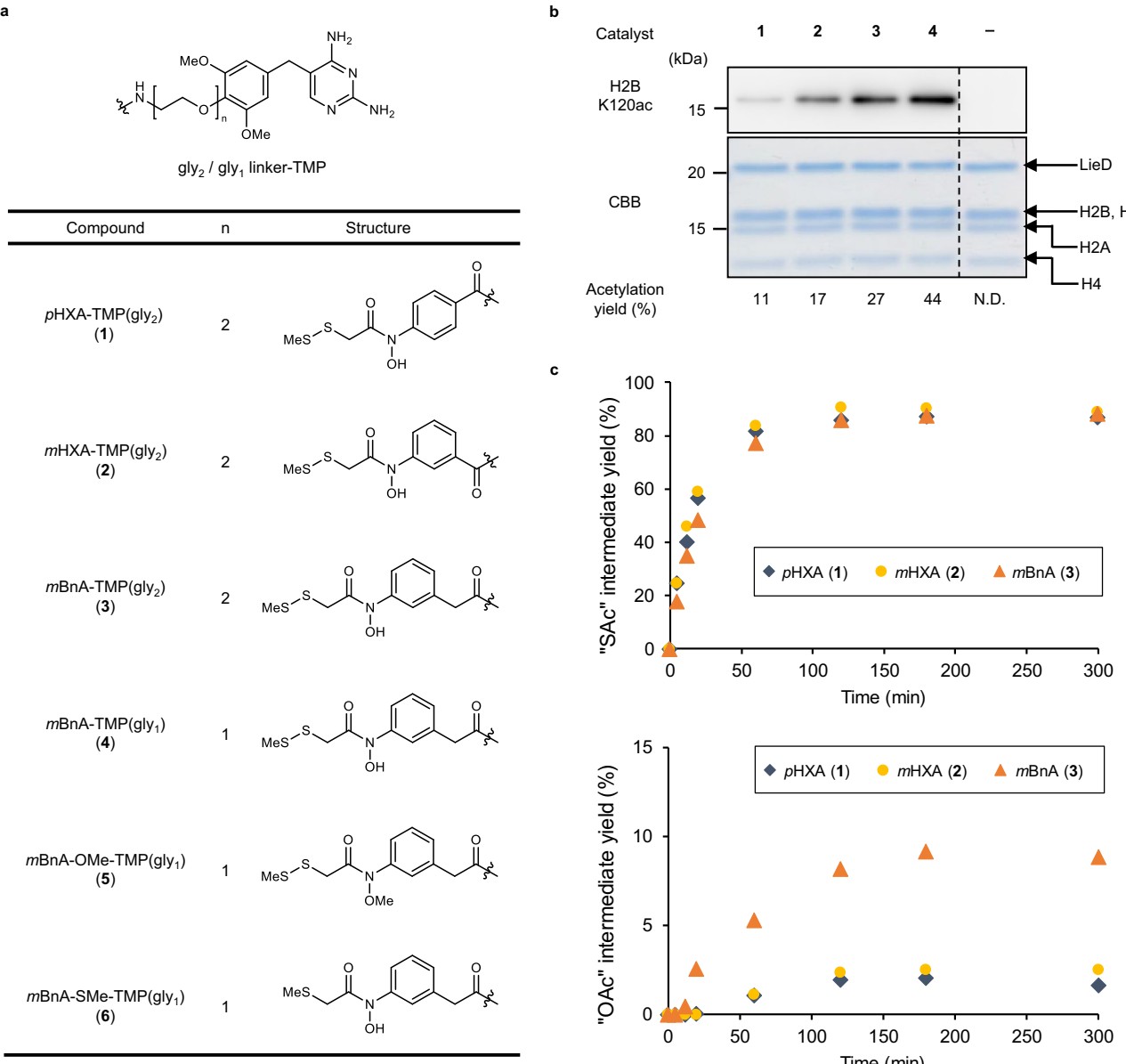

**Fig. 3 | Catalyst optimization in test tube identified *m*BnA as an active histone acetylation catalyst. a** Chemical structures of the catalysts. The *O*- and *S*-methylated compounds of *m*BnA-TMP(gly₁) (**5**, **6**) are negative control catalysts with protected nucleophilic centers. **b** Acetylation of histone H2BK120 in test tube reactions. Recombinant mono-nucleosome (0.35 μM) was reacted with each catalyst (5 μM) in the presence of Ac-CoA (1 mM), LieD protein ligand (2 μM), and TCEP (200 μM) at 37 °C for 5 h, and acetylation was detected with anti-H2BK120ac antibody by western blot analysis. Total histones and LieD proteins were visualized by CBB staining. Dividing lines have been used to indicate where noncontiguous sections of a gel have been aligned for ease of comparison. Representative data

from two independent experiments are shown. Acetylation yield at H2BK120 was quantified by LC–MS/MS analysis. The mean chemical yields of the two independent experiments are shown. "N.D." denotes "not detected". Source data are provided as a Source Data file. **c** Generation of *S*- (upper) and *O*-acetylated (lower) intermediates of the indicated catalysts. Each catalyst (25 μM) was reacted with Ac-CoA (5 mM) in the presence of TCEP (200 μM) at 37 °C for the indicated time, and the acetylated intermediates were quantified by HPLC analysis. The mean chemical yields of the two independent experiments are shown. Source data are provided as a Source Data file.

H2BK120 was below the detection limit. Treatment of LieD-expressed cells with **4**, however, resulted in an obvious enhancement of H2BK120 acetylation. This result highlights the superiority of chemical catalyst **4** over HAT enzymes in H2BK120 acetylation.

### Evidence of utilization of endogenous Ac-CoA by *m*BnA

To confirm that *m*BnA used endogenous Ac-CoA as the acetyl source in the in-cell H2BK120 acetylation reaction, we carried out metabolic isotopic labeling of Ac-CoA with [13]C-labeled glucose[29]. [U-[13]C]-glucose, which is metabolically converted into Ac-CoA with two isotopically labeled carbons in the acetyl group

(Supplementary Fig. 14), was added to cell medium not containing glucose. First, LieD-transfected HEK293T cells were treated with [U-[13]C]-glucose (4.5 g/L) for 16 h. The cells were then treated with *m*BnA-TMP **4** for 10 h, histones were purified and digested with trypsin and Glu-C peptidases, and an expected two-Dalton mass shift at H2BK120ac was detected by LC–MS/MS analysis (Fig. 4b, c). In the mass traces shown in Fig. 4c, "All Pr" denotes a non-acetylated histone peptide containing H2BK120 (note: the unreacted free lysines are propionylated in preparation for LC–MS/MS analysis: see Methods section), while "K120ac" denotes a peptide acetylated at H2BK120. Control samples with no

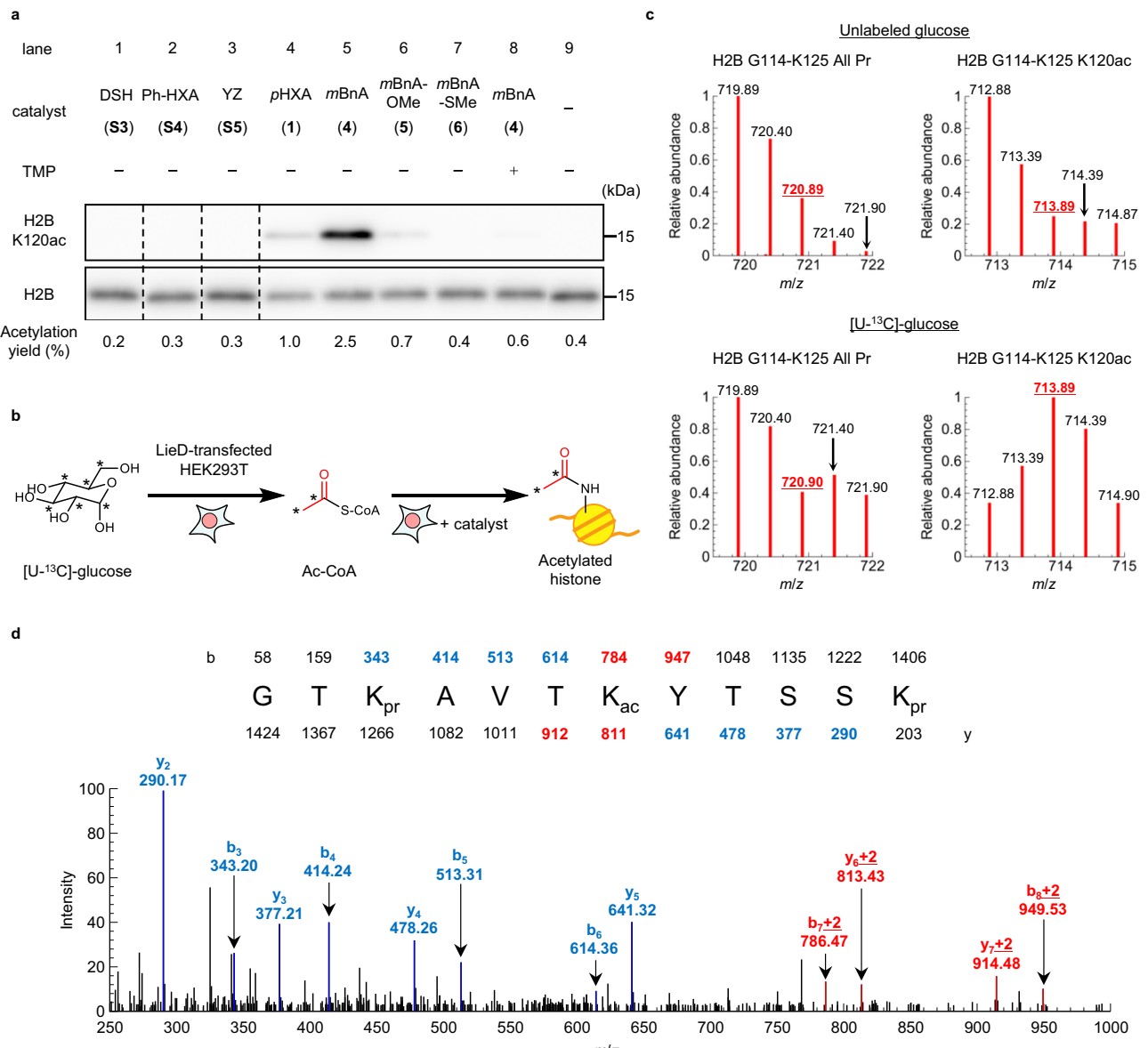

**Fig. 4 | The optimized catalyst *m*BnA can acetylate a histone lysine residue in living cells using endogenous Ac-CoA. a** Acetylation of histone H2BK120 in living cells. LieD-transfected HEK293T cells were treated with the indicated catalysts (100 μM) for 10 h. Histone proteins were acid-extracted, and H2BK120 acetylation and total H2B subunit were detected with anti-H2BK120ac and anti-H2B antibodies, respectively, by western blot analysis. Dividing lines have been used to indicate where noncontiguous sections of a gel have been aligned for ease of comparison. Representative data from two independent experiments are shown. The acetylation yields at H2BK120 were quantified by LC−MS/MS analysis. The mean chemical yields of two independent experiments are shown. Source data are provided as a Source Data file. **b** Schematic illustration of metabolic isotopic labeling of Ac-CoA using [U-13C]-glucose. The *U* refers to "universally labeled" (i.e., all carbons are 13C). An asterisk denotes the position of 13C-incorporation. **c** Detection of isotopic labeling at acetylated H2BK120. LieD-transfected HEK293T cells were incubated in a medium containing non-labeled glucose (upper) or [U-13C]-glucose (lower), and

were treated with *m*BnA-TMP(gly₁) **4**. Acid-extracted histones were treated with propionic anhydride to propionylate non-acetylated lysine residues, digested by trypsin and Glu-C peptidases, and then analyzed by LC−MS/MS. The [M + 2H]²⁺ parent peaks were analyzed. Isotopic distribution of the H2B G114-K125 peptide, whose lysine residues were all propionylated (i.e., non-acetylated, All Pr) are shown on the left, while that of the mono-acetylated peptide (K120ac) is shown on the right. As we observed [M + 2H]²⁺ peaks, +2 Da shift was detected as +1 *m/z* shift, as shown in red (720.90 and 713.89). Representative data from two independent experiments are shown. Source data are provided as a Source Data file. **d** LC−MS/MS trace of the H2B G114-K125 K120ac peptide from LieD-transfected HEK293T cells cultured in [U-13C]-glucose and treated with *m*BnA-TMP(gly₁) **4**. The sequence and the calculated *m/z* values of the fragment ions are shown. Peaks shown in red indicated incorporation of +2 Da shift at H2BK120. Representative data from two independent experiments are shown. Source data are provided as a Source Data file.

glucose-labeling provide the baseline isotopic distribution for the histone peptide, and All Pr and K120ac showed the same isotopic distribution (Fig. 4c, upper panel). When [U-13C]-glucose was used, the All Pr peptide showed a slight heavy shift of the isotopic distribution because some of the proteinogenic amino acids are biosynthesized from the [U-13C]-glucose and 13C were incorporated into the peptide backbone[30] (Fig. 4c, lower panel).

Compared to this All Pr peptide, the K120ac peptide showed a significant two-Dalton mass shift of the isotopic distribution with [U-13C]-glucose addition. Additionally, tandem mass spectrometry analysis showed that the two-Dalton mass shifts occurred specifically in the MS/MS fragment containing the acetyl group at K120 (Fig. 4d). These results indicate that the endogenous acetyl donor which *m*BnA used in the H2BK120 acetylation was Ac-CoA.

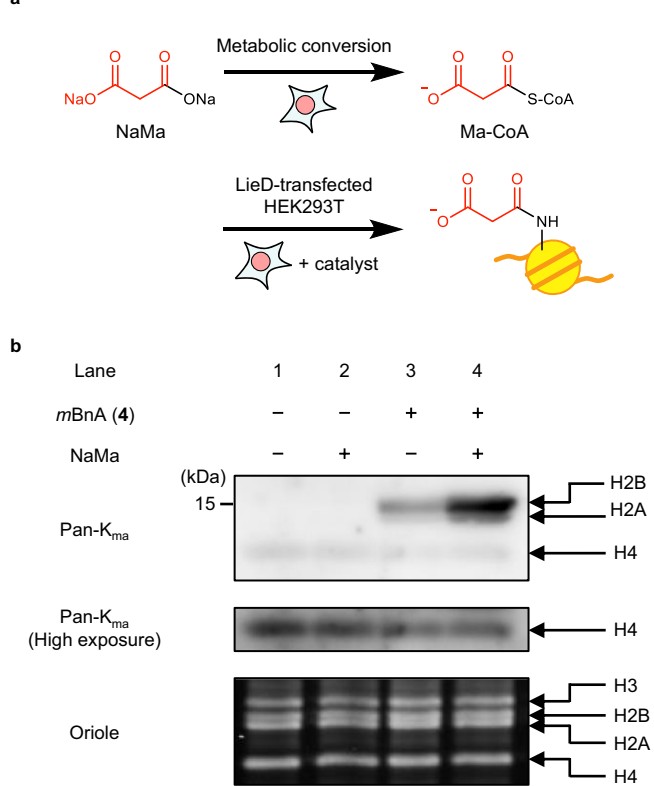

**Fig. 5 | The catalyst *m*BnA can activate endogenous non-acetyl acyl-CoA to acylate histones. a** Schematic illustration of histone malonylation. The concentration of Ma-CoA in HEK293T cells was metabolically increased by treatment with disodium malonate (NaMa). The malonyl group of Ma-CoA was then transferred to histone H2BK120 by *m*BnA catalyst. **b** Malonylation of histones in living cells. LieD-transfected HEK293T cells were treated with *m*BnA-TMP(gly₁) (**4**, 100 μM) in the presence or absence of NaMa (20 mM), and histone proteins were acid-extracted. Malonylated histones were detected by western blot analysis using pan-Kₘₐ antibody. Total histone proteins were visualized with Oriole staining. Representative data from two independent experiments are shown.

## Activation of endogenous malonyl-CoA to acylate histone

Cells contain various non-acetyl acyl-CoAs and HAT enzymes introduce various histone lysine acylation with these acyl-CoAs. However, HAT is thought to be incapable of introducing some types of acyl groups, and malonylation is one such example. Enzymatic histone malonylation has not yet been identified even though histone malonylation occurs in living cells[31–34]. Therefore, we examined whether *m*BnA could activate malonyl-CoA (Ma-CoA) to promote histone malonylation in living cells, a reaction HAT cannot catalyze. LieD-transfected HEK293T cells were treated with *m*BnA-TMP **4** for 10 h, and the histones purified from these cells were analyzed by western blot using an anti-malonyl lysine antibody. The malonylation level of histone H2B was increased in the cells treated with *m*BnA-TMP **4** (lane 1 vs. 3, Fig. 5a, b). The malonylation level of H4, which is the major endogenous malonylation target, was not affected by the *m*BnA-TMP **4** treatment, suggesting that the *m*BnA catalyst promoted the H2B malonylation not by stimulating the endogenous malonylation mechanisms, but through its own catalytic activity. Intracellular concentrations of acyl-CoAs are known to increase by treatment with the corresponding sodium carboxylates (Fig. 5a)[31,32]. Treatment with both disodium malonate (20 mM) and *m*BnA-TMP **4** further promoted the malonylation, demonstrating that the *m*BnA catalyst has the ability to activate non-acetyl, endogenous acyl-CoA to acylate histone lysine residues.

## Probing nuclear-cytosolic acyl-CoA concentrations by *m*BnA

Lastly, we investigated utilizing *m*BnA-TMP **4** to analyze subcellular acyl-CoA levels in living cells. Histone malonylation promoted by **4** was enhanced by the addition of disodium malonate, a metabolic precursor of Ma-CoA, indicating that yield of **4**-mediated histone acylation is acyl-CoA concentration-dependent (Fig. 5b, Supplementary Fig. 11, and Supplementary Fig. 15a). This was also confirmed in test-tube reactions using variable concentrations of Ac-CoA and Ma-CoA (Supplementary Fig. 6 and Supplementary Fig. 15b). In living cells, the acyl-CoAs available to the catalyst for histone acylation are present in the nucleus and any cytosol in contact with it through nuclear pores (i.e., in the nuclear-cytosolic compartment), and the effect of mitochondrial acyl-CoAs can be excluded.

Generally, the global histone acylation level is considered to be simply correlated to the concentration of cellular acyl-CoA. In some cases, however, histone acylation levels are not coupled to the whole-cell concentration of acyl-CoA[35,36]. Subcellular compartment-specific production and consumption of acyl-CoA may cause this gap, but these are still not fully understood due to the difficulty of fractionating subcellular compartments[37–39]. Because detectable H2BK120 acetylation and malonylation were catalyzed solely by catalyst **4**, not by endogenous enzymes, we envisioned **4**-mediated H2BK120 acylation as a way to probe acyl-CoA levels in the nuclear-cytosolic compartment in response to extracellular nutrient stimuli.

We investigated the effects of glucose concentration in culture media on nuclear-cytosolic acyl-CoA levels. Glucose is the major metabolic precursor of Ac-CoA, which is further converted to Ma-CoA in the fatty acid synthesis pathway (Fig. 6a). LieD-transfected HEK293T cells pre-cultured in a 1 g/L glucose-containing medium were incubated with 1 g/L or 4.5 g/L glucose for 24 h. After reaction with *m*BnA-TMP **4** for 10 h, histone acylations were analyzed by western blot (Fig. 6b). Consistent with previous reports[35,36], enzymatic acetylation at H3K9 and H3K18 increased under the higher glucose concentration, which produced a higher whole-cell Ac-CoA concentration[40]. In contrast, **4**-mediated H2BK120 acetylation decreased. This result indicates that the Ac-CoA concentration of the nuclear-cytosolic compartment decreased under high glucose conditions. On the other hand, H2BK120 malonylation increased at the higher glucose concentration, indicating that nuclear-cytosolic Ma-CoA concentration increased under the same conditions.

Distinct metabolic precursors of Ac-CoA induce variable histone acetylation[41]. We, therefore, next utilized catalyst **4** as a probe for assessing nuclear-cytosolic Ac-CoA/Ma-CoA levels upon the addition of various sodium carboxylates as their metabolic precursors (Fig. 6a, c). The addition of sodium acetate (NaAc) increased the H2BK120ac level, indicating that the nuclear-cytosolic Ac-CoA concentration increased. This response is in contrast to the response to the addition of glucose (Fig. 6b). Disodium malonate (NaMa) significantly increased both H2BK120ac and histone malonylation levels, indicating increases in nuclear-cytosolic Ac-CoA and Ma-CoA concentrations. However, NaMa only minimally affected the enzymatic histone acetylation levels at H3K9 and H3K18. Disodium succinate (NaSucc) and trisodium citrate (NaCit), known as intermediates of the tricarboxylic acid (TCA) cycle, decreased H2BK120ac and histone malonylation levels, indicating decreases in the nuclear-cytosolic Ac-CoA and Ma-CoA concentrations, without affecting enzymatic histone acetylation. Sodium butyrate (NaBu), which is simultaneously a precursor in mitochondrial Ac-CoA production and an HDAC inhibitor, potently enhanced endogenous histone acetylation, while H2BK120ac and histone malonylation, and by extension nuclear-cytosolic Ac-CoA and Ma-CoA, were significantly decreased. These results suggest that concentrations of cellular acyl-CoAs and enzymatic histone acylation are not linearly correlated. Taken together, our work demonstrated that chemical

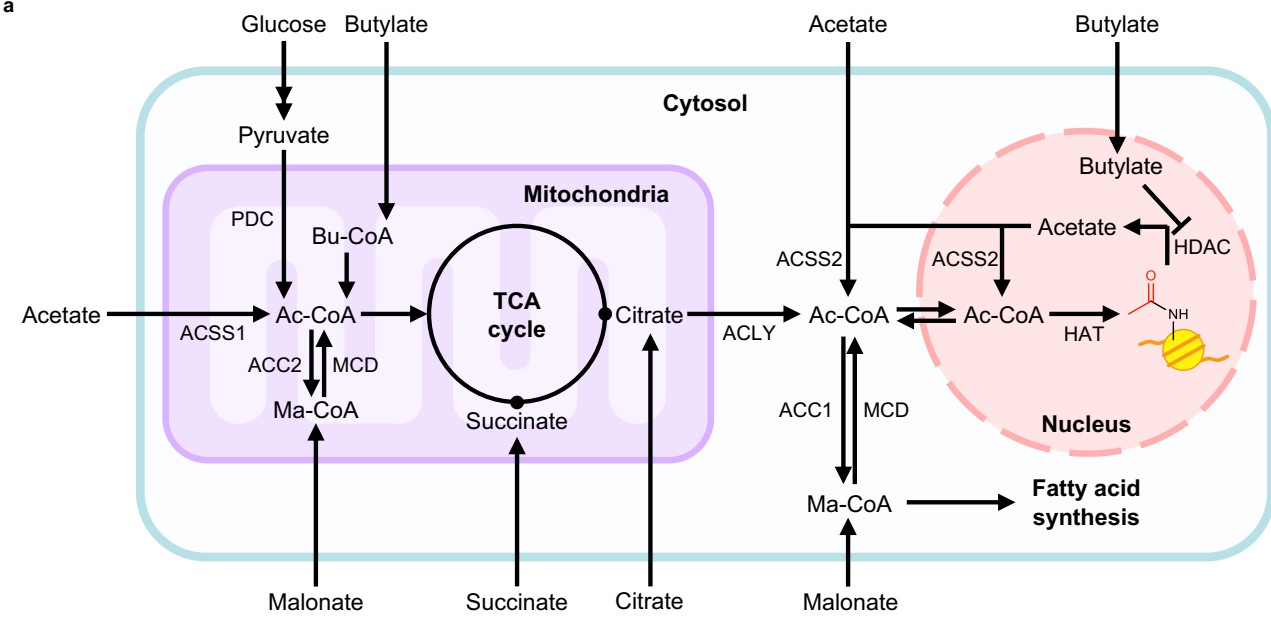

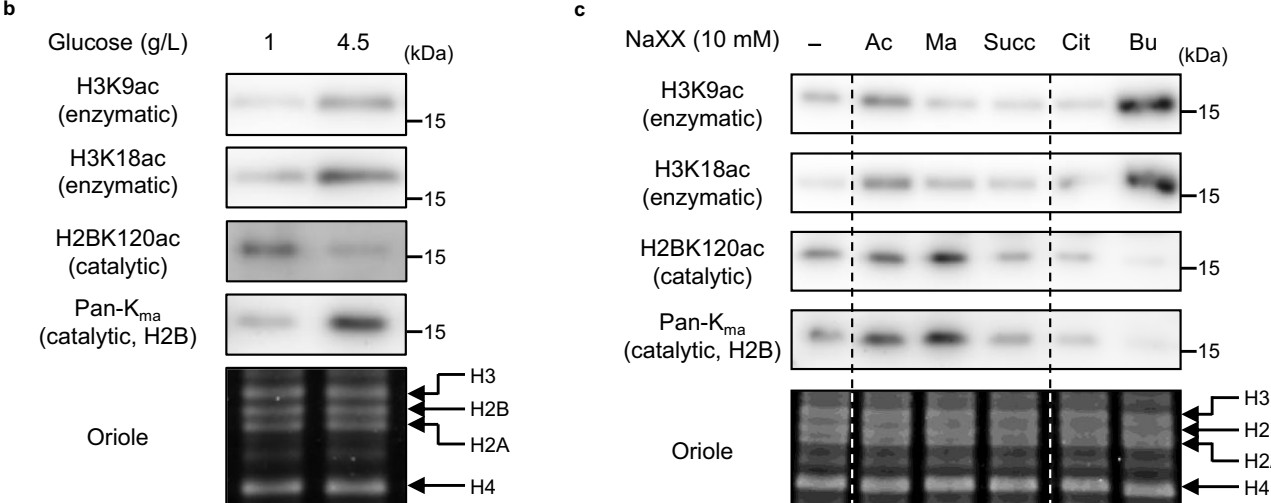

**Fig. 6 | Chemical catalyst 4 as a probe to detect changes in concentration of nuclear-cytosolic acyl-CoAs in response to metabolic perturbations.**
**a** Metabolic schematics of glucose and various carboxylates producing Ac-CoA and Ma-CoA. ACSS acetyl-CoA synthetase, PDC pyruvate dehydrogenase complex, ACC acetyl-CoA carboxylase, MCD malonyl CoA decarboxylase, ACLY ATP-citrate lyase. **b**, **c** Histone acylation levels under different glucose concentrations (**b**) or in the presence of various sodium carboxylates (NaXXs, **c**). The culture media of LieD-transfected cells containing 1 g/L glucose were replaced with the media supplemented with 10 mM sodium carboxylates. After 24-h incubation, the cells were

treated with *m*BnA-TMP(gly₁) (**4**, 100 μM) for 10 h. Acetylation at H3K9, H3K18, and H2BK120 in the whole-cell extract was detected using anti-H3K9ac, anti-H3K18ac, and anti-H2BK120ac antibodies, respectively, by western blot analysis. Histone malonylation in the whole-cell extract was detected using a pan-K$_{ma}$ antibody by western blot analysis. Total histone proteins were visualized with Oriole staining. Dividing lines have been used to indicate where noncontiguous sections of a gel have been aligned for ease of comparison. Representative data from two independent experiments are shown. Ac acetate, Ma malonate, Succ succinate, Cit citrate, Bu butylate.

catalyst **4** is a unique tool capable of evaluating changes in the concentration of nuclear-cytosolic acyl-CoAs in response to stimuli affecting acyl-CoA metabolism.

## Discussion

We developed the lysine acylation catalyst *m*BnA, which can acylate histone proteins solely with endogenous acyl-CoAs. A thiol group as an acyl-CoA-capturing motif, the hydroxamic acid as a nucleophilic catalyst core, and the suitable electronic density/acidity of the hydroxamic acid are all key to this success. To achieve lysine acylation with low concentrations of acyl-CoAs in cells, facile generation of the active acyl species (i.e., acyl hydroxamate), its sufficient reactivity with lysine

residues, and its in-cell stability circumventing non-productive pathways (e.g., hydrolysis and reactions with glutathione) must be all satisfied concurrently. The superior nucleophilicity of the hydroxamate anion is hypothesized to contribute to the facile generation of the acyl hydroxamate, and the suitable acidity of the hydroxamic acid in the *m*BnA catalyst is hypothesized to contribute to the good balance of the reactivity and stability of the acyl hydroxamate. Detailed mechanism studies illuminating structure-activity relationships of the catalysts are now underway.

Previous studies revealed that thiophenol derivatives are capable of using endogenous Ac-CoA to promote lysine acetylation of proteins in cells[25,42]. The substrate proteins are, however, not those

endogenously acetylated and thus the in-cell acetylation was not relevant to physiological processes. Furthermore, the acetylation ability largely depends on the local amino acid environment of the substrates (e.g., acetyl transfer from proximal cysteine to lysine[42] or intrinsic high reactivity[25]). Lysine residues undergoing physiological PTM regulations are often less reactive to avoid background noise[43]. We confirmed that the 2-mercaptophenylacetic acid-derived catalyst (YZ-TMP, **S5**) was not able to promote histone acetylation (Fig. 4a), probably due to the low nucleophilicity of histone H2BK120 compared to that of its reported acetylation target, androgen receptor (AR) K720 (Supplementary Fig. 16a–b). *m*BnA catalyst is an unique example of a chemical catalyst promoting physiologically important protein acylation solely with endogenous Ac-CoA in live cells.

Chemical histone acetylation using solely endogenous Ac-CoA without the addition of exogenous acetyl reagents is favorable for synthetic manipulation of the cellular epigenome. By doing away with the need for exogenous acetyl reagents, undesirable perturbations to cells, as seen in examples where some types of thioesters (and thiols formed after hydrolysis) work as histone deacetylase (HDAC) inhibitors[44,45], can be minimized. Furthermore, the utilization of endogenous Ac-CoA, which is metabolically replenished in cells, could potentially regulate acetylation levels in a sustained manner. In terms of future in vivo applications, the promotion of histone acetylation reactions using endogenous metabolite Ac-CoA is desirable, as it might be difficult to accumulate sufficient concentrations of both the catalyst and exogenous donors at the target site simultaneously.

*m*BnA catalyst was also able to activate the non-acetyl acyl-CoA, Ma-CoA, to transfer its malonyl group to a histone lysine residue. Since there is no HAT identified to promote lysine malonylation, it is difficult to regulate histone malonylation with methods relying on enzymes, despite malonylation's profound effects on nucleosome structures/interactions[16,46,47]. The ability of *m*BnA catalyst to promote malonyl transfer from Ma-CoA indicates that this chemical surrogate for HATs could not only replace but also complement the functions of HATs. It was recently reported that carboxylic acids from foods and drugs are metabolized to form non-canonical acyl-CoAs in body, and the acyl groups are enzymatically or non-enzymatically incorporated into histone lysine residues where they may induce transcriptional perturbations[48,49]. The ability of *m*BnA to acylate histones in an acyl-CoA concentration-dependent manner could be potentially valuable in studying and intervening in the effects of exogenous carboxylic acids and the resulting non-canonical protein lysine acylations in cells, including those HATs cannot promote.

We demonstrated that *m*BnA-TMP **4** can probe nuclear-cytosolic Ac-CoA and Ma-CoA levels without technically cumbersome and metabolically disruptive subcellular fractionation and mass spectrometry-based analysis[37,39,50]. This was possible due to the following three characteristics of our chemical catalyst-based intervention: (1) yield of acylation at H2BK120 promoted by **4** was tunable depending on the concentration of Ac-CoA or Ma-CoA, (2) **4** promoted H2BK120 acetylation and malonylation solely with endogenous acyl-CoA in the nuclear-cytosolic compartment, and (3) H2BK120 acetylation and malonylation are enzyme-independent. Cellular acyl-CoA metabolism is a complex interaction of multiple and varied factors that resists elucidation; using **4** to probe subcellular acyl-CoA levels has allowed us to begin teasing apart and understanding the different processes involved. The result that the level of **4**-mediated acetylation decreased under elevated glucose conditions is especially intriguing, as it was previously reported that the amount of Ac-CoA at the whole-cell level increased under high-glucose conditions[40] (Fig. 6b). This result suggests that changes in nuclear-cytosolic acyl-CoA concentrations are independent of mitochondrial acyl-CoA concentrations. Under glucose-rich conditions, glucose-derived Ac-CoA generated in mitochondria is actively transported to the nuclear-cytosolic compartment in the form of citrate, which is then utilized for histone

acetylation by HATs and for Ma-CoA production to store excess carbohydrates as fatty acids[51] (Fig. 6a). The increase in enzymatic acetylation at the histone tail region and Ma-CoA production would consume Ac-CoA, resulting in a decrease in nuclear-cytosolic Ac-CoA. In contrast, sodium acetate is directly converted to Ac-CoA by the nuclear-cytosolic isoform of acetyl-CoA synthetase 2 (ACSS2)[52], circumventing deficiency (Fig. 6c). Similarly, disodium malonate is directly converted to Ma-CoA and then to Ac-CoA in cytosol[31,53], leading to an increase in nuclear-cytosolic Ac-CoA and Ma-CoA levels. The addition of excess amounts of the TCA cycle intermediates succinate and citrate resulted in the decline of both Ac-CoA and Ma-CoA levels, though citrate can be converted to cytosolic Ac-CoA by the cytosolic enzyme ATP-citrate lyase (ACLY). However, stimulation of the TCA cycle by the addition of citrate could increase the requirement for Ac-CoA, leading to a decrease in Ac-CoA and Ma-CoA levels. An HDAC inhibitor, sodium butylate, induced a significant decrease in both Ac-CoA and Ma-CoA levels. This was likely due to inhibition of HDAC- and ACSS2-driven acetyl group recycling for Ac-CoA regeneration[54]. More importantly, our results revealed that enzymatic histone acetylation levels and nuclear-cytosolic Ac-CoA levels often do not correlate to each other, as shown in the outcomes following addition of glucose or sodium butylate. This observation suggests the possibility that histone acetylation is mainly regulated by the activity of HATs and HDACs, rather than Ac-CoA concentration. This hypothesis is consistent with previous reports suggesting HAT activity (e.g., autoacetylation to enhance substrate affinity, induction of nuclear localization, and improved stability[55–58]) has a profound impact on the regulation of histone acetylation levels[35,36]. Further investigations will provide more insight into the compartmentalized nature of the metabolism of acyl-CoAs and its influence on the various histone acylations.

The current acylation target of *m*BnA-TMP **4**, H2BK120, has potential for the chemical manipulation of the cellular epigenome. We previously reported that H2BK120 acetylation that was chemically introduced using an exogenous acetyl donor has therapeutic potential in MLL-rearranged leukemia, by blocking the downstream histone ubiquitination required for the cancer progression[15]. It was also demonstrated that acetylation and malonylation of H2BK120 attenuate internucleosomal interactions[16], suggesting that those modifications might affect higher-order chromatin structures. Nonetheless, catalysts capable of targeting other histone lysine residues, such as those in histone tail domains, are also appealing. Additionally, unlike the LieD system used in this report (which requires overexpression of an exogenous protein), direct targeting of endogenous nucleosomes in the native cellular environment is ideal. Replacing the TMP motif of **4** with other nucleosome-targeting motifs, such as octaarginine and pyrrole-imidazole-polyamide[16,47,59], is one approach to developing such a catalyst. Gene-selective histone acylation, achieved by delivering the catalyst to specific gene loci, is also important for the future realization of transcriptional regulation by HAT-surrogate catalysts.

In summary, we have developed a lysine acylation catalyst, *m*BnA, which serves as a surrogate for endogenous histone acetyltransferases by promoting acyl group transfer from cellular acyl-CoAs to histone protein lysine residues. This is a step forward in the development of a small-molecule chemical catalyst with functionality equal to or greater than endogenous enzymes, capable of intervening in the cellular chemical network.

## Methods

### Preparation of the recombinant proteins
The nucleosomes were reconstituted by the salt dialysis method using the human histone octamer and 601 nucleosome-positioning sequence DNA, and were purified as described previously[47,60]. Recombinant LieD protein was obtained as described previously[23]. Recombinant AR protein (551–919) was purchased from abcam (ab235857).

### Antibodies

Rabbit polyclonal antibodies against H2B (abcam, ab1790, 1:2000 dilution for WB), H3K9ac (Merck, 07-352, 1:1000 dilution for WB), H3K18ac (abcam, ab1181, 1:1000 dilution for WB), and pan-$K_{ma}$ (PTM-Biolabs, PTM-901, 1:1000 dilution for WB), a rabbit monoclonal antibody against p300 (Cell Signaling Technology, 54062, 1:1000 dilution for WB), and mouse monoclonal antibodies against H2BK120ac (1 µg/mL for WB) and H3K27ac (Merck, 05-1334, 1:1000 dilution for WB) were used for western blot and ChIP assay. The monoclonal antibody against H2BK120ac was generated as described previously[15]. Secondary antibodies, anti-rabbit HRP (Cell Signaling Technology, 7074S, 1:5000 for WB) and anti-mouse HRP (GE Healthcare Life Sciences, NA931V, 1:10000 for WB or Jackson ImmunoResearch, 515-035-003, 1:10000 for WB) were also used for western blot. Normal rabbit IgG (SantaCruz, sc-2027) and Normal mouse IgG (SantaCruz, sc-2025) were also used for ChIP assay. The amounts of antibodies used for ChIP assay were indicated in Supplementary Methods section.

### General method for protein digestion for LC−MS/MS analysis

The propionylated protein samples were resuspended in 50 mM NH$_4$HCO$_3$ aq. with 0.1% ProteaseMAX (Promega, V2072) and digested by the digestive enzymes indicated in Supplementary Table 1–2, 10 ng/µL Trypsin Gold (Promega, V5280) with or without 10 ng/µL Glu-C (Promega, V1651) or 2 ng/µL Asp-N in 50 mM NH$_4$HCO$_3$ aq. with 0.02% ProteaseMAX at 37 °C for 3 h or overnight. Then 5% aqueous formic acid (vol/vol) was added, and the solvents were removed by Speed-Vac evaporator to obtain dried, digested samples, which were dissolved in 0.1% aqueous formic acid (vol/vol). After centrifugation (21,130 g, 10 min), the supernatant was used for LC−MS/MS analysis.

### LC−MS/MS analysis

LC−MS/MS analysis was carried out using a TripleTOF 5600$^+$ instrument equipped with eksigent M5 microLC (AB Sciex). LC was carried out as follows: 3C18-CL-120 column (0.5 mm ID × 100 mm) using 2% acetonitrile with 0.1% formic acid for 1 min, followed by a linear gradient of 2-35% acetonitrile with 0.1% formic acid over 6 min, and then a linear gradient of 35-90% acetonitrile with 0.1% formic acid over 1 min at 40 °C with a flow rate of 20 µL/min. The volume of injected samples was 5 µL. The eluent was monitored by on-line quadrupole time-of-flight mass spectrometer (ESI-Q-TOF MS), operated in positive ion mode. The targeted precursor ions and fragment ions, and collision energy for each peptide are shown in Supplementary Table 1–2. Data analysis was carried out using PeakView version 1.2 (AB Sciex) and the averaged result or the representative result obtained from duplicate measurements of each two different experiments is reported as the result for quantification of acylation yields or detection of isotope labeling respectively. The yield of acylated lysine was calculated from the extracted ion chromatogram as a percentage of the peak area for acylated peptides divided by the combined peak areas for both acylated and propionylated peptides. The MS and MS/MS spectrum chart was obtained as an impulse plot of peak heights.

### Histone acylation in test tube

A catalyst-TMP (5 µM) was mixed with TCEP (200 µM) in Tris buffer (50 mM Tris-HCl (pH 7.5), 100 mM NaCl) and the mixture was incubated at 37 °C for 5 min before recombinant nucleosomes (0.35 µM), recombinant LieD proteins (2 µM), and the indicated concentration of Ac-CoA or Ma-CoA (1 mM unless otherwise stated) were added to the solution. The mixture was incubated at 37 °C for 5 h (or 10 h in the experiment for Supplementary Fig. 6 and Supplementary Fig. 15a). For immunoblotting, the reaction mixture was boiled with SDS sample buffer and analyzed by CBB staining and western blot using the anti-H2BK120ac or pan-$K_{ma}$ antibody. Representative data of two independent experiments are shown. For LC−MS/MS analysis, proteins in the reaction mixture were precipitated by trichloroacetic acid (TCA,

16.6%). The protein was collected by centrifugation, air-dried, and dissolved in Milli-Q water (MQ). After DNA was digested by DNase I (Takara, 2270 A) for 30 min at 37 °C, the samples were mixed with acetone (74%) and incubated overnight at −30 °C, then the proteins were collected by centrifugation, air-dried, and dissolved in MQ. To the solution, 50 mM aqueous ammonium bicarbonate (NH$_4$HCO$_3$ aq.) and 25% propionic anhydride solution (methanol/propionic anhydride, 3:1 (vol/vol)) were added, and pH was adjusted to 8 by adding ammonia solution. After 1 h incubation at r.t., the solvents were removed by Speed-Vac evaporator. The propionylated proteins were treated as described in "General method for protein digestion for LC−MS/MS analysis" section.

### HPLC analysis for the formation of the *S*- and *O*-acetylated intermediates of the catalysts

A catalyst-TMP conjugate (25 µM) was mixed with TCEP (200 µM) in Tris buffer (50 mM Tris-HCl (pH 7.5), 100 mM NaCl), and the mixture was incubated at 37 °C for 5 min before Ac-CoA (5 mM) was added to the solution. The mixture was incubated at 37 °C for the indicated time. At each time point, the reaction was stopped by adding 0.5% TFA aqueous solution (final 0.4%), and was analyzed with HPLC and LC/MS. The yields of the *S*- and *O*-acetylated intermediates were calculated according to the equation below. Since *N-O* bond of hydroxamic acids was possibly reduced with TCEP, peak area of "SH" was calculated as the sum of hydroxamic acid-SH and reduced amide-SH. The same calculation method was applied to "SAc".

$$\text{``SAc'' intermediate yield (\%)} = \frac{\left(Area_{\text{``SAc''}} + Area_{\text{``SAc,OAc''}}\right)}{\left(Area_{\text{``SH''}} + Area_{\text{``SAc''}} + Area_{\text{``OAc''}} + Area_{\text{``SAc,OAc''}}\right)} \times 100$$

(1)

$$\text{``OAc'' intermediate yield (\%)} = \frac{\left(Area_{\text{``OAc''}} + Area_{\text{``SAc,OAc''}}\right)}{\left(Area_{\text{``SH''}} + Area_{\text{``SAc''}} + Area_{\text{``OAc''}} + Area_{\text{``SAc,OAc''}}\right)} \times 100$$

(2)

### General methods for in-cell experiments

HEK293T cells (kindly provided by the Toru Hirota laboratory) were incubated at 37 °C and 5% CO$_2$. DMEM++ refers to Dulbecco's Modified Eagle Medium (DMEM, Gibco, 11965092) supplemented with 10% fetal bovine serum (FBS) and GlutaMAX. DMEM+++ refers to DMEM++ supplemented with penicillin and streptomycin unless otherwise stated. CRB buffer refers to 50 mM Tris-HCl (pH 7.5), 300 mM NaCl, 0.3% Triton X-100. CRB++++ buffer refers to CRB buffer supplemented with MgCl$_2$ (2 mM), PMSF (1 mM), protease inhibitor cocktail (Sigma-Ardrich, P2714-1BTL), and benzonase nuclease (Novagen, 0.1 U/mL). CRB+++++ buffer refers to CRB ++++ buffer supplemented with sodium butylate (5 mM).

### In-cell acetylation of histones using endogenous Ac-CoA

In all, 2.8 × 10$^5$ HEK293T cells in 1 mL DMEM++ were seeded into wells of a 12-well plate. After 24 h incubation, the cells were transfected with pcDNA5/TO-LieD@R52-FLAG plasmid (1.25 µg per well; plasmid constructed as reported previously[23]) using Lipofectamine LTX and Plus reagents according to the manufacturer's instructions. After 8 h, the medium was carefully removed and replaced with 1 mL DMEM+++ supplemented with BSO (100 µM). After 16 h, the medium was carefully removed and replaced with 0.4 mL Opti-MEM (Gibco, 31985070) supplemented with catalyst-TMP (100 µM unless otherwise stated), BSO (100 µM), and 0.5% DMSO. After 10 h (or the indicated time), the cells were detached by pipetting vigorously, transferred to an Eppendorf tube, centrifuged (304 g, 5 min, 4 °C), and washed with PBS. In all, 420 µL Extraction Buffer of Histone Purification Kit (Active Motif) were added to cells and then the mixture was incubated with agitation

overnight at 4 °C. After centrifugation (21,130×g, 10 min), the supernatant was collected. For immunoblotting, the histones were precipitated by TCA (25%). The proteins were collected by centrifugation, washed by acetone, air-dried, and dissolved in MQ. Collected proteins were boiled with SDS sample buffer and analyzed by western blot using the indicated antibodies. The samples derived from the same experiment and the gels/blots were processed in parallel. Representative data of two independent experiments are shown. For LC–MS/MS analysis, histone H2A/H2B and H3/H4 subunits were isolated separately with the Histone Purification Kit (Active Motif) according to the manufacturer's instructions. In brief, the H2A/H2B and H3/H4 fractions were isolated from the acid-extracted histone solution by subsequent elution. The histone subunits in each fraction were precipitated with perchloric acid (final 6%). The precipitates were washed with 4% perchloric acid twice, 2% HCl/acetone twice, and acetone twice, followed by air-drying. The histones were dissolved in 20 μL MQ, then 20 μL of 200 mM $NH_4HCO_3$ aq. and 40 μL of propionic anhydride solution was added, and pH was adjusted to 8 by adding ammonia solution. After 30 min incubation at r.t., the solvents were removed by Speed-Vac evaporator. The propionylated proteins were treated as described in "General method for protein digestion for LC–MS/MS analysis" section.

### In-cell acetylation of histones with $^{13}C$ isotopic labeling

In all, $2.8 \times 10^5$ HEK293T cells in 1 mL high glucose DMEM++ (Gibco, 11965) were seeded into wells of a 12-well plate. After 24 h incubation, the cells were transfected with pcDNA5/TO-LieD@R52-FLAG plasmid (1.25 μg per well) using Lipofectamine LTX and Plus reagents according to the manufacturer's instructions. After 8 h, the medium was carefully removed and replaced with 1 mL high-glucose DMEM+++ supplemented with BSO (100 μM) or no-glucose DMEM+++ (Gibco, 11966) supplemented with [U-$^{13}C$]-glucose (4.5 g/L) and BSO (100 μM). After 16 h, the medium was carefully removed and replaced with 0.4 mL high-glucose or no-glucose DMEM+++ supplemented with catalyst-TMP (100 μM), [U-$^{13}C$]-glucose (4.5 g/L for no-glucose DMEM+++), BSO (100 μM), and 0.5% DMSO. After 10 h, the cells were detached by pipetting vigorously, transferred to an Eppendorf tube, centrifuged (304 g, 5 min, 4 °C), and washed with PBS. For LC–MS/MS analysis, histone H2A/H2B and H3/H4 subunits were isolated separately using the Histone Purification Kit (Active Motif) according to the manufacturer's instructions. The histone subunits in each fraction were dissolved in 20 μL MQ, then 20 μL of 200 mM $NH_4HCO_3$ aq. and 40 μL of propionic anhydride solution were added, and pH was adjusted to 8 by adding ammonia solution. After 30 min incubation at r.t., the solvents were removed by Speed-Vac evaporator. The propionylated proteins were treated as described in "General method for protein digestion for LC–MS/MS analysis" section.

### In-cell malonylation of histones using endogenous Ma-CoA

In all, $2.8 \times 10^5$ HEK293T cells in 1 mL no-glucose DMEM++ supplemented with glucose (1 g/L) were seeded into wells of a 12-well plate. After 24 h incubation, the cells were transfected with pcDNA5/TO-LieD@R52-FLAG plasmid (1.25 μg per well) using Lipofectamine LTX and Plus reagents according to the manufacturer's instructions. After 12 h, the medium was carefully removed and replaced with 1 mL DMEM +++ supplemented with glucose (1 g/L) and BSO (100 μM, only the experiment for Fig. 5b) with or without disodium malonate (NaMa, 20 mM unless otherwise stated). After 24 h, the medium was carefully removed and replaced with 0.4 mL no-glucose DMEM +++ supplemented with catalyst-TMP (100 μM), glucose (1 g/L), BSO (100 μM), NaMa (20 mM), and 0.5% DMSO, and the mixture was incubated for 10 h. For immunoblotting, the histones were collected from the cells as described in "In-cell acetylation of histones using endogenous Ac-CoA"

section and boiled with SDS sample buffer. The proteins were analyzed by Oriole staining and western blot using the pan-$K_{ma}$ antibody. For immunoblotting of whole-cell extracts, the cells were lysed with CRB++++ buffer on ice for 30 min, and then treated as described in "In-cell acetylation of histone with p300/CBP siRNA treatment" section. Representative data of two independent experiments are shown.

### Investigating changes in nuclear-cytosolic acyl-CoA levels through in-cell acylation of histones

In all, $2.8 \times 10^5$ HEK293T cells in 1 mL no-glucose DMEM++ supplemented with glucose (1 g/L) were seeded into wells of a 12-well plate. After 24 h incubation, the cells were transfected with pcDNA5/TO-LieD@R52-FLAG plasmid (1.25 μg per well) using Lipofectamine LTX and Plus reagents according to the manufacturer's instructions. After 12 h, the medium was carefully removed and replaced with 1 mL DMEM+++ supplemented with the indicated concentration of glucose (1 g/L unless otherwise stated) with or without the indicated sodium carboxylates (NaXXs, 10 mM). After 24 h, the medium was carefully removed and replaced with 0.4 mL no-glucose DMEM+++ supplemented with catalyst-TMP (100 μM), glucose, NaXX, and 0.5% DMSO, and the mixture was incubated for 10 h. For immunoblotting of whole-cell extracts, the cells were lysed with CRB++++ buffer on ice for 30 min, and then treated as described in "In-cell acetylation of histone with p300/CBP siRNA treatment" section. Representative data of two or three independent experiments are shown.

### Reporting summary

Further information on research design is available in the Nature Portfolio Reporting Summary linked to this article.

## Data availability

All data supporting the conclusions and findings included in this study are available within the article or the Supplementary Information/ Source Data file. Uncropped and unprocessed scans of all blots and gels are provided in the Source Data file. Source data are provided with this paper.

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

## Acknowledgements

We thank J. Kato (Univ. Tokyo) for her assistance. We thank R. Newlon (Univ. Tokyo) for reading the manuscript and providing helpful comments. We thank JSPS KAKENHI Grant Numbers JP20H00489 (M.K.), JP23H05466 (M.K.), JP19KK0179 (S.A.K., K.Y., and M.K.), JP21K19326 (K.Y.), JP22H05018 (K.Y.), JP21H02074 (S.A.K.), JP18H05534 (H.Ku.), JP23H05475 (H.Ku.), JST ERATO grant number JPMJER1901 (H.Ku.), and SUNBOR GRANT (K.Y.) for financial support. This research was partially supported by Research Support Project for Life Science and Drug Discovery (Basis for Supporting Innovative Drug Discovery and Life Science Research (BINDS)) from AMED under Grant Number JP 22ama121009j0001 (H.Ku.).

## Author contributions

S.M., S.A.K., K.Y., and M.K. conceived and designed the project. M.H., S.M., and H.Ka. performed the experiments and synthesized all chemical compounds. T.K. and H.Ku. purified the recombinant mononucleosomes. M.H., S.A.K., K.Y., and M.K. wrote the manuscript with contributions from all the other authors.

## Competing interests

The authors declare no competing interests.
