## [Peer Review File · Nature Communications]

Reviewers' Comments:

Reviewer #1:

Remarks to the Author:

The manuscript by Habazaki et al. describes the development of the small-molecule chemical catalyst mBnA that enables the upregulation of histone acetylation with endogenous acyl-CoA. The hydroxamic acid-based mBnA is a surrogate for the histone acetyltransferase (HAT) and enables the activation of endogenous acyl-CoAs, thus promoting histone lysine acylation in living cells without needing to add exogenous acyl donors. Besides an upregulation of histone acetylation, mBnA also increases the levels of cellular histone malonylation, which is a PTM that is not catalyzed by HAT. Therefore, the authors provide a highly interesting new concept that cannot only replace but also complement the functions of HAT. In my opinion, mBnA will be an important tool for future synthetic intervention in the cellular epigenome, and studying the effects of histone acylation under physiological and pathological conditions.

I therefore suggest this manuscript to be published in Nature Communications. However, the following (minor) points should be addressed prior to publication.

- Figure 3a: For clarity, compounds 5-6 should be labeled as negative control catalysts, either in the table or in the legend.
- Figure 3b: For western blot quantification, the number of replicates and statistical errors have to be provided.
- Figure 4a: For western blot quantification, the number of replicates and statistical errors have to be provided.
- In the discussion, the authors should address the fact that they only used their catalyst in a non-native/artificial cellular environment (i.e. LieD-transfected HEK293T).
- Figure S1: "Chemical structure of Ph-HXA, which was (not: is) previously reported"

Reviewer #2:

Remarks to the Author:

In this manuscript, Habazaki et al reported their fantastic work regarding the development of a chemical catalyst motivating histone acylation (e.g., H2B K120Ac) with endogenous acyl-CoA. Overall, the manuscript is well written, and the conclusions are clearly presented with the strong support of well-organized figures and comprehensive data. The discovery and validation of this novel chemical catalyst for site-specific histone acylation expand the chemical tool box for epigenetics studies. I believe that after the publication of this paper, a number of labs will reach out to the authors asking for this fascinating molecule to pursue their chemical epigenetics research (maybe including my own lab). Thus, I highly recommend the publication of this paper on Nat. Commun. after minor revisions:

1. The authors utilized LieD to guide their chemical catalyst for the site-specific acylation of H2B K120. Could they give another application example of this molecule to acylate an N-terminal Lys residue of any core histones? Many of the histone N-terminal Lys acylations have more biological significance than H2B K120Ac.
2. To exclude the enzymatic effects, the authors performed two experiments: competition assay with free TMP and detection of H3K9Ac and H3K18Ac catalyzed by HATs. However, more direct evidence should be provided. For example, the specific inhibition of H2B K120Ac writer, KAT3 (using either pharmacologic inhibition or RNAi), should be performed in the presence and absence of chemical catalyst.
3. The epigenetics impacts of this chemical catalyst should be detected. For example, the crosstalk between H2B K120Ub and chemical catalyst-caused K120Ac should be determined. The overall chromatin changes induced by chemical catalyst should be detected using ATAC-Seq and RNA-Seq. More importantly, ChIP-Seq of H2B K120Ac needs to be conducted to show the specific gene regions modified by the chemical catalyst in the whole genome in comparison with endogenous enzymatic ones (mediated by KAT3).

Reviewer #3:

Remarks to the Author:

In this manuscript, the authors develop a small molecule catalyst, mBnA, and demonstrate its capacity

to promote the transfer of an acyl group from endogenous Acyl-CoA to histone H2B K120. Overall, this is an interesting study, however, the concept of activating Acyl-CoA with small molecules is not new. In fact, the authors (and others) have published several manuscripts detailing similar systems recently. In addition, I have some significant concerns about the claims of the utility of this approach without supporting evidence. Given this precedence of similar systems and the fact the authors do not provide additional application of this strategy or further understanding/improvement of substrate scope, selectivity and/or efficiency, this current work represents more of a technical vignette rather than a substantial advancement suitable for publication in *Nature Communications*. Therefore, I believe this work better suited for publication in a more specialized journal. I provide more detail of this conclusion below.

- The authors very recently reported LANA-DSH, a thiol containing DMAP compound conjugated to a chromatin targeting LANA peptide and demonstrated its ability to catalyze acetyl transfer to H2BK10 in living cells and suppress ubiquitination (*PNAS*, 2021, e2019554118). The authors also published (*Chem. Asian J.* 15, 833–839, 2020) an improved piperidine-conjugated hydroxamic acid (Ph-HXA) and other DSH analogs for live cell acylation chemistries (*J. Am. Chem. Soc.* 143, 14976–14980 (2021); *ACS Chem. Biol.* 2019, 14, 6, 1102–1109). In addition, as the authors point out, other groups have also developed similar small molecule surrogates for lysine acetyltransferases (*ACS Chem. Biol.* 2016, 11, 10, 2797–2802; *Nat. Chem. Biol.* 2010, 6, 887–889). In this manuscript, the authors generate additional analogs based on the DSH and Ph-HXA, and once again demonstrate its ability to activate acetyl-CoA and induce acetylation of H2B K120. The authors do not provide any advanced applications or understanding of these reagents as synthetic HATs.
- The main impact that the authors highlight continuously throughout the manuscript is the ability to use such synthetic HAT systems to study acetylation in live cells. However, they do not demonstrate such applications supporting this claim. Also, I have fundamental concerns with this claim in general. The authors only monitor acetylation of H2BK120, H3K9 and H3K18. However, by activating Acyl-CoA for nucleophilic attack by lysines in an enzyme-independent fashion using 100uM mBnA, I would suspect that many, many other proteins (both histones and non-histones) are also getting acylated throughout the proteome. The authors provide no additional characterization beyond H3K9 and H3K18, nor provide any evidence of tunability or controllability, which I suspect there is none given the mechanism of general acyl-CoA activation. Together, this makes it difficult to understand how such systems (at least in their current form) would be useful to investigate acetylation in an endogenous setting.
- The efficiency of this catalyst has not been completely characterized. Working concentration range of Acetyl-CoA is not tested in vitro. This data is important since a major cellular obstacle is the low concentration of Acyl CoA. In the cellular experiment, the authors incubated the catalyst with cells for 10 hours. Acetylation and deacetylation often occur at much faster time scales. Once again the broader effects of long term incubations of high concentrations of mBnA is not provided. Further, given the kinetics of acetylation, more rapid acetylation would likely be useful in biologically relevant settings.

RESPONSES TO REVIEWERS' COMMENTS:

Reviewer #1 (Remarks to the Author):

The manuscript by Habazaki et al. describes the development of the small-molecule chemical catalyst mBnA that enables the upregulation of histone acetylation with endogenous acyl-CoA. The hydroxamic acid-based mBnA is a surrogate for the histone acetyltransferase (HAT) and enables the activation of endogenous acyl-CoAs, thus promoting histone lysine acetylation in living cells without needing to add exogenous acyl donors. Besides an upregulation of histone acetylation, mBnA also increases the levels of cellular histone malonylation, which is a PTM that is not catalyzed by HAT. Therefore, the authors provide a highly interesting new concept that cannot only replace but also complement the functions of HAT. In my opinion, mBnA will be an important tool for future synthetic intervention in the cellular epigenome, and studying the effects of histone acylation under physiological and pathological conditions.

I therefore suggest this manuscript to be published in Nature Communications. However, the following (minor) points should be addressed prior to publication.

We appreciate that you evaluate our chemical catalyst to be an important tool for future synthetic intervention in the cellular epigenome as a HAT surrogate. We have addressed all the points that you kindly brought up. A point-by-point response to your comments is attached below.

- Figure 3a: For clarity, compounds 5-6 should be labeled as negative control catalysts, either in the table or in the legend.

Thank you for your comment. As you have suggested, we have labeled compounds **5–6** as negative control catalysts in the legend of **Fig. 3a**.

- Figure 3b: For western blot quantification, the number of replicates and statistical errors have to be provided.

- Figure 4a: For western blot quantification, the number of replicates and statistical errors have to be provided.

We appreciate these helpful suggestions. We have conducted two independent experiments

for all western blot analyses for histone acylation, and have presented the representative data. We have described those points in the legend of all western blot analysis data. In **Fig. 3b, 4a**, and **Supplementary Fig. 5**, the acetylation yields at H2BK120 were quantified by LC-MS/MS analysis (not western blot quantification). We have clarified this point in the legends, and added the statistical errors of the two independent quantifications to the figures.

- In the discussion, the authors should address the fact that they only used their catalyst in a non-native/artificial cellular environment (i.e. LieD-transfected HEK293T).

Thank you for your important suggestion. We clarified this important point in the Discussion section (page 28, line 16-18) as follows: *“Additionally, unlike the LieD system used in this report (i.e., needing overexpression of an exogenous protein), direct targeting of endogenous nucleosomes in a native cellular environment would be ideal.”* We hope you find our revised description acceptable.

- Figure S1: “Chemical structure of Ph-HXA, which was (not: is) previously reported”

We corrected as you suggested.

Reviewer #2 (Remarks to the Author):

In this manuscript, Habazaki et al reported their fantastic work regarding the development of a chemical catalyst motivating histone acylation (e.g., H2B K120Ac) with endogenous acyl-CoA. Overall, the manuscript is well written, and the conclusions are clearly presented with the strong support of well-organized figures and comprehensive data. The discovery and validation of this novel chemical catalyst for site-specific histone acylation expand the chemical tool box for epigenetics studies. I believe that after the publication of this paper, a number of labs will reach out to the authors asking for this fascinating molecule to pursue their chemical epigenetics research (maybe including my own lab). Thus, I highly recommend the publication of this paper on *Nat. Commun.* after minor revisions:

Thank you for carefully reading our manuscript. We are glad that you appreciate our catalyst *mBnA* as a fascinating molecule to pursue chemical epigenetic studies. We have addressed all the points that you kindly commented. A point-by-point response to your comments is attached below.

1. The authors utilized LieD to guide their chemical catalyst for the site-specific acylation of H2B K120. Could they give another application example of this molecule to acylate an N-terminal Lys residue of any core histones? Many of the histone N-terminal Lys acylations have more biological significance than H2B K120Ac.

Thank you for your important comment. We understand that the histone N-terminal Lys residues are attractive acylation targets due to their biologically significant effects on the epigenome. Targeting histone N-terminal Lys residues is, however, currently difficult because of the lack of histone-binding molecules for recruiting the catalyst to a specific N-terminal Lys residue. As you insightfully commented, acylation of histone N-terminal Lys residues is an attractive and important future challenge for us. This point was described in the Discussion section (from page 28, line 12).

2. To exclude the enzymatic effects, the authors performed two experiments: competition assay with free TMP and detection of H3K9Ac and H3K18Ac catalyzed by HATs. However, more direct evidence should be provided. For example, the specific inhibition of H2B K120Ac writer, KAT3 (using either pharmacologic inhibition or RNAi), should be performed in the presence and absence of chemical catalyst.

Thank you for the comment. As suggested, we have conducted the acetylation reaction by *mBnA* catalyst with inhibition of KAT3 by RNAi. The results indicate *mBnA* was still able to promote H2BK120 acetylation. This clearly excludes the possibility that the observed H2BK120 acetylation was mediated by KAT3, which is the writer enzyme of H2BK120ac (Gatta *et al. Epigenetics* **2011**, 6, 630–637). We added these data in the new **Supplementary Fig. 10**, and added the following sentences on page 16, line 3-5: “Furthermore, **4** promoted acetylation of H2BK120 when H2BK120ac writer enzymes p300 (KAT3B) and CBP (KAT3A) were knocked down with RNAi (**Supplementary Fig. 10**).”

Additionally, we performed a more global quantitative analysis of histone lysine acetylation encompassing more than just H3K9ac and H3K18ac by LC-MS/MS. Acetylation levels at lysine residues other than those proximal to the catalyst binding site were not affected by the catalyst treatment. This result also supports the hypothesis that the observed histone acetylation was promoted by the catalyst, not by cellular enzymes. We presented these data in the new **Supplementary Fig. 9** in place of the original figure showing the western blot analysis of H3K9 and H3K18 acetylation. We also added the following sentence on page 16, line 2-3: “Acetylation levels at histone lysine residues other than those proximal to the LANA binding site did not change by treatment with *mBnA*-TMP **4** (**Supplementary Fig. 9**).”

3. The epigenetics impacts of this chemical catalyst should be detected. For example, the crosstalk between H2B K120Ub and chemical catalyst-caused K120Ac should be determined. The overall chromatin changes induced by chemical catalyst should be detected using ATAC-Seq and RNA-Seq. More importantly, ChIP-Seq of H2B K120Ac needs to be conducted to show the specific gene regions modified by the chemical catalyst in the whole genome in comparison with endogenous enzymatic ones (mediated by KAT3).

Thank you for the insightful comments. Based on your suggestion, we performed ChIP-qPCR analysis to elucidate possible gene region dependency of our H2BK120 acetylation. We found that H2BK120 in both the euchromatic and heterochromatic regions were similarly acetylated by *mBnA* catalyst. This result indicates that our catalyst acylates histones with consistent efficiency irrespective of the chromatin structure and the transcriptional state. We added these data to **Supplementary Fig. 12** and described the results on page 16, line 9-13.

It was reported that H2BK120ac appears as an early mark of gene activation (Gatta *et al. Epigenetics* **2011**, 6, 630–637). We previously reported that acetylation of H2BK120

attenuates internucleosomal interactions (Amamoto *et al. J. Am. Chem. Soc.* **2017**, *139*, 7568–7576) and chemically introduced H2BK120 acetylation blocks enzyme-dependent ubiquitination of the same lysine residue (Fujiwara *et al. PNAS* **2021**, *118*, e 2019554118). These results indicate that H2BK120 acetylation introduced by *mBnA* in this study would exhibit epigenetic impacts. We could not observe them at the current stage however, likely due to the modest acetylation yield. The current work is critical in showcasing the concept of the HAT-surrogate small-molecule chemical catalyst that can activate endogenous Ac-CoA in living cells. The investigation of the epigenetic impact resulting from this approach is an important future challenge. Instead, we have demonstrated that the catalyst is a novel tool for monitoring intracellular Ac-CoA levels (see below in Responses to Reviewer 3). This new result has been included in the revised manuscript.

Reviewer #3 (Remarks to the Author):

In this manuscript, the authors develop a small molecule catalyst, mBnA, and demonstrate its capacity to promote the transfer of an acyl group from endogenous Acyl-CoA to histone H2B K120. Overall, this is an interesting study, however, the concept of activating Acyl-CoA with small molecules is not new. In fact, the authors (and others) have published several manuscripts detailing similar systems recently. In addition, I have some significant concerns about the claims of the utility of this approach without supporting evidence. Given this precedence of similar systems and the fact the authors do not provide additional application of this strategy or further understanding/improvement of substrate scope, selectivity and/or efficiency, this current work represents more of a technical vignette rather than a substantial advancement suitable for publication in Nature Communications. Therefore, I believe this work better suited for publication in a more specialized journal. I provide more detail of this conclusion below.

We appreciate your invaluable comments. Based on your comments and suggestions, we have addressed your concerns regarding the utility and properties of our catalyst in the revised manuscript. The revised version of the manuscript has been significantly strengthened owing to your valuable comments. We hope that you find our revised manuscript suitable for publication in Nature Communications as reviewers #1 and #2 have.

- The authors very recently reported LANA-DSH, a thiol containing DMAP compound conjugated to a chromatin targeting LANA peptide and demonstrated its ability to catalyze acetyl transfer to H2BK120 in living cells and suppress ubiquitination (PNAS, 2021, e2019554118). The authors also published (Chem. Asian J. 15, 833–839, 2020) an improved piperidine-conjugated hydroxamic acid (Ph-HXA) and other DSH analogs for live cell acylation chemistries (J. Am. Chem. Soc. 143, 14976–14980 (2021); ACS Chem. Biol. 2019, 14, 6, 1102–1109). In addition, as the authors point out, other groups have also developed similar small molecule surrogates for lysine acetyltransferases (ACS Chem. Biol. 2016, 11, 10, 2797–2802; Nat. Chem. Biol. 2010, 6, 887–889). In this manuscript, the authors generate additional analogs based on the DSH and Ph-HXA, and once again demonstrate its ability to activate acetyl-CoA and induce acetylation of H2B K120. The authors do not provide any advanced applications or understanding of these reagents as synthetic HATs.

As you mentioned, we previously reported several catalysts (e.g., DSH and Ph-HXA) for

lysine acetylation of proteins, including histones with low reactivity. Although those catalysts activate a high concentration of Ac-CoA and promote selective protein acetylation **in test tubes**, **all of the reported catalysts that are functional in cells required addition of exogenous, cell-permeable, non-Ac-CoA acetyl donors**. Those previous catalysts were **NOT capable of acetylating histone with in-cell endogenous Ac-CoA as the sole acetyl donor**.

Some thiophenol derivatives have been reported to promote in-cell protein lysine acetylation with endogenous Ac-CoA (Zhang *et al. ACS Chem. Biol.* **2016**, *11*, 2797–2802; Jenkins *et al. Nat. Chem. Biol.* **2010**, *6*, 887–889). However, the target lysine residues have been restricted to exceptionally reactive ones. For example, the reacting lysine residue in androgen receptor AR (AR K720: reported in Zhang *et al. ACS Chem. Biol.*) possesses decreased pK_a due to the existence of basic amino acid residues nearby. Acetylation of such reactive lysine residues is physiologically not functional. On the other hand, lysine residues undergoing physiologically significant modifications (e.g., epigenetically important histone proteins) are of low reactivity (Hacker *et al. Nat. Chem.* **2017**, *9*, 1181–1190). To clarify the novelty of our present work, we compared the reactivities of AR K720 and histone H2BK120 to find that H2BK120 was of ca. 3-fold lower reactivity than AR K720 (new **Supplementary Fig. 16b**). Furthermore, the thiophenol-derived catalyst YZ, previously used for acetylation of AR K720, was not able to promote histone acetylation (**Fig. 4a**)

The *mBnA* catalyst reported in this study is therefore the first chemical catalyst enabling lysine acetylation of **epigenetically important but low-reactivity histones solely using endogenous Ac-CoA**. Furthermore, *mBnA* catalyst can promote histone malonylation with in-cell Ma-CoA, which is not achievable with enzymes or catalysts previously reported by either ourselves or others. These facts demonstrate the novelty of *mBnA* catalyst reported in this study.

• The main impact that the authors highlight continuously throughout the manuscript is the ability to use such synthetic HAT systems to study acetylation in live cells. However, they do not demonstrate such applications supporting this claim.

We appreciate your interest in seeing what synthetic HAT systems can achieve. In the revised manuscript, we have additionally investigated the ability of the catalyst *mBnA* to detect nuclear-cytosolic Ac-CoA/Ma-CoA levels under various nutrient conditions. Using the ability of *mBnA* to promote H2BK120 acetylation/malonylation depending on the cellular Ac/Ma-CoA

concentrations, we discovered varying effects of different nutrient conditions on cellular Ac/Ma-CoA concentrations. As a result, we obtained novel insight into the relationships between nuclear-cytosolic acyl-CoA concentrations altered by different metabolic perturbations and histone lysine acetylation levels. Important findings are: 1) Glucose and sodium acetate have contrasting effects on the nuclear-cytosolic Ac-CoA concentration despite both nutrients being reported to increase the Ac-CoA level in the whole cell. This observation suggests that the *mBnA* catalyst is a useful probe for determining the local Ac-CoA level in the nuclear-cytosolic compartment in living cells. 2) Enzymatic histone acetylation seems often disconnected from free Ac-CoA concentration under some nutrient conditions. This suggests the possibility that histone acetylation is regulated largely dependent on the activity of HATs and HDACs rather than on the Ac-CoA concentration. These important insights have been obtained for the first time because an in-cell acyl-CoA-sensing catalyst capable of such was developed in this study. Please see the details in the new subsection <**Probing nuclear-cytosolic acyl-CoA concentrations by *mBnA***> in the Results section (from page 20) and the Discussion section (from page 26, line 15). We greatly appreciate that our revised manuscript has been significantly strengthened thanks to your insightful comments.

- Also, I have fundamental concerns with this claim in general. The authors only monitor acetylation of H2BK120, H3K9 and H3K18. However, by activating Acyl-CoA for nucleophilic attack by lysines in an enzyme-independent fashion using 100uM *mBnA*, I would suspect that many, many other proteins (both histones and non-histones) are also getting acylated throughout the proteome.

Thank you for your comments. To investigate whether acylation of other non-histone proteins was promoted by the *mBnA* catalyst, we visualized lysine malonylation in whole-cell extract of the catalyst-treated cells by western blot using a pan-malonyl lysine antibody (**Supplementary Fig. 11**, note: we chose malonylation rather than acetylation to more clearly see the effect of catalyst-dependent acylation without high background signals from endogenously acetylated proteins in cells). As a result, promotion of lysine malonylation of non-histone proteins were not observed, confirming high histone-selectivity of *mBnA*-mediated lysine acylation.

We also quantified acetylation levels of global histone lysine residues by LC-MS/MS analysis in the presence and absence of *mBnA* catalyst to investigate the residue selectivity of the

histone acetylation. The results showed that *mBnA* catalyst promoted acetylation only at the lysine residues proximal to its binding site on the nucleosome (i.e., H2BK116 and H2BK120 with 1:4 selectivity), and acetylation levels of other lysine residues were not affected by the catalyst treatment (new **Supplementary Fig. 9** in place of the original figure monitoring the acetylation of H2BK120, H3K9, and H3K18 by western blot). We have described this point in page 16, line 2-3.

- The authors provide no additional characterization beyond H3K9 and H3K18, nor provide any evidence of tunability or controllability, which I suspect there is none given the mechanism of general acyl-CoA activation. Together, this makes it difficult to understand how such systems (at least in their current form) would be useful to investigate acetylation in an endogenous setting.

In addition to the characterization of *mBnA*-mediated acetylation selectivity (new **Supplementary Fig. 9** and see above), we have also addressed the tunability and controllability of the catalyst-mediated acetylation according to your helpful comment. The *mBnA*-mediated H2BK120 acetylation level in cells is tunable depending on the reaction time and catalyst concentration, as shown in the new **Supplementary Fig. 8**. Furthermore, the *mBnA*-mediated H2BK120 acetylation level is tunable by the concentration of acyl-CoA (new **Supplementary Fig. 6** and **Supplementary Fig. 15**). This acyl-CoA concentration-dependency of H2BK120 acetylation led to the application of the *mBnA* catalyst as a chemical probe for evaluating nuclear-cytosolic acyl-CoA levels as described above. We appreciate your attention regarding the practicality this system.

- The efficiency of this catalyst has not been completely characterized. Working concentration range of Acetyl-CoA is not tested *in vitro*. This data is important since a major cellular obstacle is the low concentration of Acyl CoA.

In response to your suggestion, we have investigated the working concentration range of Ac-CoA in our catalytic system *in vitro*. Under conditions with concentrations of Ac-CoA below 50 μM , which corresponds to the concentration range of cellular Ac-CoA, the *mBnA* catalyst-mediated H2BK120 acetylation yield showed clear Ac-CoA concentration-dependency. Histone acetylation was promoted with as low as 1 μM Ac-CoA. We included this result in the new **Supplementary Fig. 6** with the following description on page 13, line 5-8: “The concentration of endogenous Ac-CoA in cells is reported to be around 3–50 μM . *mBnA*-TMP

4 promoted H2BK120-selective histone acetylation in test tubes with less than 50 μ M Ac-CoA, suggesting that 4 can activate endogenous Ac-CoA and promote histone acetylation in living cells (Supplementary Fig. 6)."

In the cellular experiment, the authors incubated the catalyst with cells for 10 hours. Acetylation and deacetylation often occur at much faster time scales. Once again the broader effects of long term incubations of high concentrations of mBnA is not provided. Further, given the kinetics of acetylation, more rapid acetylation would likely be useful in biologically relevant settings.

Thank you for your insightful comment. As explained above, the 10 hour incubation of cells with mBnA catalyst did not alter acetylation levels of non-targeted lysine residues of cellular proteins (new **Supplementary Fig. 9** and **Supplementary Fig. 11**), which indicates that broader unintended effects of the long term incubations are minimum. As for the reaction kinetics, we conducted time-course experiments of mBnA-mediated histone acetylation and found that acetylation already proceeded to some extent at a shorter reaction time of 1-2 h (new **Supplementary Fig. 8a**). The turnover rate of histone lysine acylation varies by lysine residue, and that of H2BK120ac is especially slow as H2BK120ac is not the substrate of histone deacetylase (HDAC), as demonstrated in our previous report (Fujiwara *et al. PNAS* **2021**, *118*, e2019554118, see Figure 4B and S5). For a lysine residue with a slow acetylation turnover rate, such as H2BK120, mBnA-mediated acylation with cellular acyl-CoA serves as a useful probe to investigate the subcellular acyl-CoA status as exemplified in the various acyl-CoA nutrient experiments described above and in **Fig. 6**. As you insightfully mentioned, lysine acetylation of histone tail domains has a higher turnover rate. For application to histone tail domains, the catalyst activity will need to be further improved. This will be an important challenge of our future research.

REVIEWERS' COMMENTS

Reviewer #1 (Remarks to the Author):

The manuscript by Habazaki et al. describes the development of the small-molecule chemical catalyst mBnA that enables the upregulation of histone acetylation with endogenous acyl-CoA. The hydroxamic acid-based mBnA is a surrogate for the histone acetyltransferase (HAT) and enables the activation of endogenous acyl-CoAs, thus promoting histone lysine acylation in living cells without needing to add exogenous acyl donors. Besides an upregulation of histone acetylation, mBnA also increases the levels of cellular histone malonylation, which is a PTM that is not catalyzed by HAT. Further, the developed chemical catalyst can be applied as a probe to detect changes in concentration of nuclear-cytosolic acyl-CoAs in response to metabolic perturbations.

Therefore, the authors provide a highly interesting new concept that cannot only replace but also complement the functions of HAT. In my opinion, mBnA will be an important tool for future synthetic intervention in the cellular epigenome, and studying the effects of histone acylation under physiological and pathological conditions.

In the revised version of the manuscript, the authors have addressed all my concerns to my satisfaction. I also have to say that I find the revised manuscript overall much better than the previous version. I recommend publication as it is.

Reviewer #2 (Remarks to the Author):

The authors have well addressed my concerns. I suggest the publication of this paper on Nature Communications.

Reviewer #4 (Remarks to the Author):

In this manuscript, Habazaki et al described a lysine acylation catalyst mBnA, which can catalyze the acetylation of histones using endogenous acetyl-CoA. This is an interesting work that has potential as both a therapy and as a way to elucidate functions of the epigenome for intervening in the chemical network of epigenetic regulation. The authors' work in recent years has been focused on the identification of small molecule activators (Amamoto et al., *J. Am. Chem. Soc.*, 2017) and extended our understanding in the chemical regulation of epigenetics. Altogether, this study reports a new catalyst that not only replaces but also complements the functions of HAT. I support the publication of this manuscript.

Some suggestions:

1. mBnA has a more efficient acetylation catalytic function than other activators, but whether other histone sites also undergo acetylation?
2. Whether these acetylations may affect the chromatin structure and what the possible physiological consequence it has? The authors may discuss possibilities.
3. mBnA activates Acyl-CoA for nucleophilic attack by lysines in an enzyme-independent fashion with a dose of 100uM, what about the low concentration of mBnA, will it still functional ?

RESPONSES TO REVIEWERS' COMMENTS:

Reviewer #1 (Remarks to the Author):

Reviewer's comment

The manuscript by Habazaki et al. describes the development of the small-molecule chemical catalyst mBnA that enables the upregulation of histone acetylation with endogenous acyl-CoA. The hydroxamic acid-based mBnA is a surrogate for the histone acetyltransferase (HAT) and enables the activation of endogenous acyl-CoAs, thus promoting histone lysine acylation in living cells without needing to add exogenous acyl donors. Besides an upregulation of histone acetylation, mBnA also increases the levels of cellular histone malonylation, which is a PTM that is not catalyzed by HAT. Further, the developed chemical catalyst can be applied as a probe to detect changes in concentration of nuclear-cytosolic acyl-CoAs in response to metabolic perturbations.

Therefore, the authors provide a highly interesting new concept that cannot only replace but also complement the functions of HAT. In my opinion, mBnA will be an important tool for future synthetic intervention in the cellular epigenome, and studying the effects of histone acylation under physiological and pathological conditions.

In the revised version of the manuscript, the authors have addressed all my concerns to my satisfaction. I also have to say that I find the revised manuscript overall much better than the previous version. I recommend publication as it is.

Our response

Thank you for your second-round review of our manuscript, carefully reading the revised section. We appreciate your high evaluation of our chemical catalyst as an important tool for future synthetic intervention in the cellular epigenome.

Reviewer #2 (Remarks to the Author):

Reviewer's comment

The authors have well addressed my concerns. I suggest the publication of this paper on Nature Communications.

Our response

Thank you for reviewing our manuscript again and recommending it for publication. We are pleased that our revised manuscript has addressed your concerns.

Reviewer #4 (Remarks to the Author):

Reviewer's comment

In this manuscript, Habazaki et al described a lysine acylation catalyst *mBnA*, which can catalyze the acetylation of histones using endogenous acetyl-CoA. This is an interesting work that has potential as both a therapy and as a way to elucidate functions of the epigenome for intervening in the chemical network of epigenetic regulation. The authors' work in recent years has been focused on the identification of small molecule activators (Amamoto et al., J. Am. Chem. Soc, 2017) and extended our understanding in the chemical regulation of epigenetics. Altogether, this study reports a new catalyst that not only replaces but also complements the functions of HAT. I support the publication of this manuscript.

Our response

Thank you for reviewing our manuscript and kindly providing your comments. We appreciate that you evaluated the potential of our chemical catalyst, *mBnA*, both as a therapeutic tool and as a means to better understand the functions of the epigenome in intervening within the chemical network of epigenetic regulation. A point-by-point response to your comments is attached below.

Some suggestions:

1. *mBnA* has a more efficient acetylation catalytic function than other activators, but whether other histone sites also undergo acetylation?

Our response

As you commented, selective acetylation of the target histone lysine residue without affecting PTMs at other lysine residues is important. Thus, we comprehensively examined changes in the histone acetylation levels using LC-MS/MS, and the result was already included in Supplementary Figure 9 in the previous submission. *mBnA* acetylated H2BK120 without changing the acetylation levels of most of other histone lysine residues, although nearby lysine residues undergo slight acetylation (less than one-fourth of the acetylation level promoted at H2BK120). We also described this result in the main text (page 9, line 3), as follows: Acetylation levels at histone lysine residues other than those proximal to the LANA binding site did not change by treatment with *mBnA*-TMP 4 (Supplementary Fig. 9).

2. Whether these acetylations may affect the chromatin structure and what the possible physiological consequence it has? The authors may discuss possibilities.

Our response

Thank you for your insightful suggestion. We previously demonstrated that acetylation and malonylation of H2BK120 led to a reduction of internucleosomal interactions (Amamoto *et al. J. Am. Chem. Soc.* **2017**, *139*, 7568–7576). Consequently, our chemical catalyst-mediated acetylation would modulate higher-order chromatin structures. We have added the discussion about this point on page 18.

3. mBnA activates Acyl-CoA for nucleophilic attack by lysines in an enzyme-independent fashion with a dose of 100uM, what about the low concentration of mBnA, will it still functional?

Our response

To determine the operational concentration range of *mBnA* for the acetylation of histones in cells, we conducted *in-vitro* reactions using various concentrations of *mBnA*. As a result, H2BK120 acetylation was promoted with as low as 5 μ M *mBnA*. This result was already included in Supplementary Figure 8 in the previous submission.